# Financial Knowledge and Private Health Insurance: Does Age Matter?

**DOI:** 10.3390/healthcare11202738

**Published:** 2023-10-14

**Authors:** Inmaculada Aguiar-Díaz, María Victoria Ruiz-Mallorquí

**Affiliations:** Department of Financial Economics and Accounting, Faculty of Economy, Business and Tourism, University of Las Palmas de Gran Canaria, 35001 Las Palmas, Spain; inmaculada.aguiar@ulpgc.es

**Keywords:** private health insurance, financial knowledge, financial literacy, age, older

## Abstract

(1) Background: The paper focuses on the relationship between financial knowledge (FK) and holding private health insurance (PHI), and also focuses on the effect of age on the aforementioned relationship. (2) Method: The study was carried out on a sample of 8055 individuals taken from the 2016 Financial Competences Survey (the only one available), prepared by the Bank of Spain. Unlike previous studies that limited themselves to considering numeracy as a proxy for FK, in this study, two levels of FK—basic and advanced—are considered. (3) Results: The results indicate that a higher level of FK, specifically advanced FK, increases the probability of an individual holding PHI. Regarding age, it has been observed that the relationship between FK and PHI is only relevant in middle and older age, but not in younger and adults. Therefore, it is appropriate to differentiate between basic and advanced FK, and we confirm that age exerts a moderating effect on the influence of FK on PHI. (4) Conclusion: We conclude that FK—specifically, advanced FK for middle-aged and older people—is relevant to the likelihood of an individual holding PHI, which can improve health and financial wellbeing.

## 1. Introduction

Economic literature considers knowledge as an intangible asset called human capital, which allows the productivity of companies to increase, and in turn, contributes to the economic growth of a country. In addition, according to Hussain et al. [1], healthy individuals are expected to contribute “more to production, boost productivity and have a beneficial impact on human capital formation” (p. 108). These authors also assert that the government should pay more attention to human capital by dedicating more resources to the education and health sectors. Likewise, at the individual level, knowledge, and in particular financial knowledge, is a valuable resource that helps to improve people’s quality of life. One of the most important factors in quality of life is health, which is why having health insurance represents an important decision, not only with respect to health but also financially.

In this context, the objective of this work is to analyse the relationship between financial knowledge (hereafter, FK)—the main component of financial literacy (hereafter, FL)—and holding private health insurance (hereafter, PHI), with special reference to age. The relationship between both topics stems from the fact that the decision to purchase a health insurance policy is similar to other financial decisions (e.g., [2]). According to Meyer [3], healthcare decisions have both health-related and financial implications; thus, these decisions should be based on some degree of FL. Like most financial decisions, the decision to buy insurance in the individual market requires an assessment of the uncertain costs of that purchase [4]. Moreover, health insurance is a financial product that not only helps to assure access to acute and preventive medical care, but also reduces one’s exposure to financial loss derived from future and uncertain medical expenses (e.g., [5]).

The Organisation for Economic Co-operation and Development (OECD) distinguishes between public and private health insurance on the basis of the source of funding [6]. PHI is often characterised as voluntary in contrast to public health insurance, which is compulsory and publicly funded and administered [7]. In most European countries, like Spain, major health shocks are covered by the public healthcare system, but individuals may still face costs (co-payments) for particular services that are not covered by the public scheme (e.g., dental care, medical appliances, glasses, the choice of better or faster inpatient care) [8]. In countries with a national health service (NHS), PHI plays an increasingly important role, described as ‘‘double coverage’’, for which individuals pay to access both public and private services, and choose one of these depending on the type and quality of healthcare they require [9].

FL can be defined as “the ability of an individual to obtain, understand and evaluate the relevant information for making decisions, being aware of the financial consequences” [10] (p. 31). As we have indicated, FK is the main component of FL, and many studies exclusively associate it with financial literacy (e.g., [11,12,13,14,15,16]). In addition, some authors (e.g., [17,18,19,20]) distinguish two levels of FK, namely basic and advanced, depending on the complexity of the financial concepts. For example, numeracy is basic knowledge, while understanding the “risk/return” relationship may be considered advanced financial knowledge. This distinction is important because not only is relevant the quantitative level of FK, but also the qualitative one. In this sense, Meyer [3] indicates that due to the increased financial concerns of patients, they need to know the relationship between FL and how healthcare decisions are made, especially health insurance. Along the same line, Harnett [21] asserts: “Deficits in health insurance and financial literacy correlate with poor health outcomes and increased expenditures for patients” (p. 168). Thus, the first hypothesis predicts that a higher level of FK—basic and advanced—increases the likelihood of holding PHI.

Regarding age, on the one hand, Lusardi et al. [22] argue that FL changes over a person’s lifetime. Specifically, FL is critical among certain groups, such as elderly people [23]. On the other hand, “out-of-pocket payments” may create a barrier to healthcare system utilisation, which is particularly crucial in the oldest population segments [8]. In this vein, Costa-Font and Font-Vilalta [24] explain that age is a key factor affecting how likely some people are to want private healthcare, but because insurance companies calculate premiums based on age and screen their clients for risks, they may be excluding some people—for example, the elderly—from private coverage due to adverse selection problems. In this sense, age is an important variable that distinguishes both FK and healthcare. Thus, we propose a second hypotheses that age has a moderating effect in the relationship between FK—basic and advanced—and PHI.

To our knowledge, the studies that have analysed the relationship between financial literacy or financial knowledge in the possession of PHI have referred exclusively to the USA, which does not have universal health coverage. Furthermore, previous studies have been based on samples of older people and have been limited to considering numeracy as the only component of financial knowledge. The present work aims to cover these gaps, using a sample of individuals of different ages belonging to a country with universal health coverage, as well as considering different levels of financial knowledge.

This work was carried out using a sample of individuals in Spain—this being an ideal country for this analysis for several reasons. Firstly, there is universal NHS coverage, which allows citizens access to free public services, albeit sometimes with long waiting lists. This means that Spaniards tend to take out private health insurance to speed up certain treatments or medical consultations. Jofre-Bonet [25] found that there is a positive relationship between a lower quality of public healthcare (longer waiting times) and a higher probability of holding PHI in Spain. Secondly, in Spain, the demographic increase in the number of elderly people has been significant due to the increase in life expectancy. Thirdly, in Spain, older people are supported by a public pension system and an NHS. Finally, our empirical analysis is based on the data from the Financial Competences Survey (hereafter, ECF for its Spanish acronym), carried out by the Bank of Spain. To our knowledge, in the European context, data on these variables are separately available in several household surveys, but they are treated together only in the ECF. Hence, the ECF makes it possible to measure financial knowledge and other issues related to the possession of financial products, such as insurance, and specifically health insurance. For this reason, the ECF is an ideal source of data for our purpose.

This work has a double contribution. On the one hand, to our knowledge, it is the first work that considers the influence of both basic and advanced knowledge on the probability of holding PHI. Among the studies that have analysed the determinants of holding PHI, only Fang et al. [26], Greene and Peters [27] and Paccagnella et al. [8] have considered FK. However, all these studies are limited to numeracy, a basic financial concept. It is possible that the reason is that health surveys do not contain data on FK, and household finance surveys do not include information about financial competences. Moreover, previous works are limited to middle and older age in Europe [8], and over 65 years of age in the USA [26]. In this work, we consider middle and old age, but also younger adults, which allows us to compare both age segments.

The results indicate that a higher level of FK, specifically advanced FK, increases the likelihood of holding PHI. Likewise, those results confirm that age exerts a moderating effect on the influence of FK on getting PHI. We conclude that FK is relevant for holding PHI; specifically, advanced FK for middle-aged and older people in order to improve their health and financial wellbeing.

The work is structured as follows: in the Section 2, in order to contextualise the study, we make a brief summary of PHI in Europe and Spain. In the third, we approach the theoretical foundations and set out the hypotheses. The methodology is explained in the Section 4. In the fifth we present the results. In the sixth, we discuss the results. Finally, we draw our conclusions in the Section 7.

## 2. Private Health Insurance in Europe and Spain

The OECD [28] identifies three categories of PHI: complementary, supplementary and duplicate. Complementary PHI provides full or partial funding for services that are not freely provided by the NHS. Supplementary PHI provides full or partial coverage for services that are excluded by national health insurance (e.g., dental care). Duplicate PHI covers goods and services already included under statutory health insurance by increasing consumer choice and access to different health services, for example improved quality of care and faster access to treatment.

PHI is highly differentiated among European countries, perhaps due to differences in their public healthcare systems and in the characteristics of available PHI policies (in terms of covered services and costs), as well as differences in household characteristics that affect the decision to hold such policies [8]. These authors found different combinations of the three types of additional PHI in European countries. In Austria, Belgium, Denmark, Germany and Sweden, PHI is both supplementary and complementary. In Greece, it is both duplicate and supplementary national health insurance. In Italy and Spain, PHI can duplicate, supplement or complement NHS coverage.

Many Europeans have seen their exposure to “out-of-pocket medical expenditure risk” increase because they have experienced cost-containment measures, including greater rationing of publicly provided healthcare services, increased cost sharing and greater private-sector insurance provision [29]. In addition, in most of Europe, the main health-related costs are covered by the NHS, although citizens may have to face expenses that are only partially covered (co-payments) or not covered at all by the public service—e.g., dental care, medical appliances, glasses, the choice of better or faster inpatient care, etc. [8].

In Spain, which has an almost universal NHS, covering 99.1% of the population [30], there is a combination of types of PHI with relatively low premiums [31]. PHI is usually acquired to avoid waiting lists for public NHS services. According to the Spanish Health Survey [32], 14.63% of the population has mixed health coverage (public and private). The insured group is made up of individuals who take out private policies and civil servants (representing around 1.6 million). In Spain, civil servants can choose between public and private health, through agreements between the Public Administration mutual (MUFACE, ISFAS, Mugeju), with insurers financed with public funds. According to the 2018 Healthcare Barometer [33], 78% of the individuals with double coverage declared that their main reason for purchasing PHI was long waiting lists. Because of long waiting lists, Spaniards resort to obtaining PHI in order to receive faster medical assistance, which improves their quality of life.

## 3. Literature Review

The theoretical framework of this study is based on two fields of literature, namely those of financial literacy and healthcare, specifically health insurance. The relationship between both topics stems from the fact that the decision to purchase health insurance is similar to a financial decision. In this vein, Kim et al. [34] asserts that “the selection of health insurance is a complicated decision with financial implications” (p. 4), and O’Connor and Kabadayi [35] claim that the purchase of health insurance is a financial choice.

On the one hand, the literature offers several definitions of FL with different nuances. In this work, we base our approach on the proposal by Mason and Wilson [10], which defines FL as “an individual’s ability to obtain, understand and evaluate the relevant information necessary to make decisions with an awareness of the likely financial consequences” (p. 31). Thus, FL plays an important role in consumer decision-making, due to the need for people to make fully informed financial decisions. FK is the main component of FL. In this sense, many studies exclusively associate FL with FK (e.g., [11,12,13,14,15,16].

On the other hand, the primary purpose of health insurance is to cover the expenses derived from the use of private health services, since unexpected medical expenses are the main source of background risk [36]. In this sense, there are an increasing number of works that, as a consequence of the financial concerns for patients facing out-of-pocket expenses, analyse the relationship between financial literacy and decision-making [3]. Along the same line, Harnett [21] (p. 168) asserts that patients have to carry out some investigation in order to choose the most appropriate health insurance policy for their needs, as these insurance policies are becoming more and more complex. In the literature on health insurance, some authors use the concept of “health insurance literacy” (HIL) (e.g., [2,35]). The Health Insurance Literacy Expert Roundtable [37] defines HIL as “the degree to which individuals have the knowledge, ability, and confidence to find and evaluate information about healthcare plans, select the best plan for their own (or their family’s) financial and health circumstances, and use the plan once enrolled’’ (p. 7). As we can see, HIL can be considered as an application of the term FL to the field of health.

In the following subsections, we present the arguments that show the relationship of FL, specifically FK, with PHI, as well as the effect of age in this relationship.

### 3.1. Financial Knowledge and PHI

When making health insurance decisions, consumers face problems similar to those they face when making other types of financial decisions. Both financial services and the health insurance market have experienced increasing complexity, requiring a higher level of consumer insight [2]. In the same vein, Paez et al. [38] assert that “health insurance is one of the most complex, costly, confusing products consumers purchase and use in their lifetime” (p. 226). Like most financial decisions, the decision to purchase a health insurance policy in the individual market requires consumers to consider numerous possibilities (e.g., the level of resources available to meet future medical care needs), as well as assess the costs of purchasing insurance (premiums, co-payments, opportunity costs, etc.) [4].

Fang et al. [26] asserts that financial skills are essential for navigating the health insurance market for two reasons: firstly, for their effect on individuals’ ability to evaluate the costs and benefits of purchasing a health insurance; secondly, for their effect on search costs. Thus, given the price dispersion in health insurance policies offered by different insurers, those consumers with better financial skills may be able to obtain lower prices and thus be more likely to purchase. Along the same line, Harnett [21] asserts that individuals with low financial literacy are less likely to have health insurance. According to O´Connor and Kabadayi [35], choosing and using health insurance requires an understanding of cost-sharing responsibilities (consumer’s share of medical costs they pay out-of-pocket after insurance); hence, consumers need to have basic financial skills to estimate their deductibles, coinsurance and co-payments.

The aforementioned arguments support the similarity between health insurance and other financial products, which allows the relationship between possessing FK and holding PHI to be considered. Therefore, it is important to understand how FL influences consumer behaviour [3]. To that end, previous studies suggest that low levels of financial literacy can result not only in poor understanding of healthcare costs and patient financial responsibility, but also in engagement in poor financial behaviours as they pertain to healthcare [3].

Regarding empirical evidence, Fang et al. [26], Greene and Peters [27] and Paccagnella et al. [8] found that financial numeracy is important when it comes to making a decision about PHI. However, some financial service professionals have argued that individuals must have not only basic reading and math literacy to navigate today’s world, but also basic money-management skills or some level of financial literacy [39]. In the same vein, Harnett [21] asserts that numeracy alone is not sufficient as FK. Therefore, following other previous studies [17,18,19,20], we consider that when studying the relationship between possessing FK and holding PHI, a distinction must also be made between at least two levels of FK—basic and advanced.

Thus, according to the arguments and empirical evidence, the first hypothesis is as follows:

**H1**.*A higher level of FK increases the likelihood of holding PHI*.

**H1a**.*A higher level of basic FK increases the likelihood of holding PHI*.

**H1b**.*A higher level of advanced FK increases the likelihood of holding PHI*.

### 3.2. Financial Knowledge, Age and PHI

Age has been considered in studies that have analysed the determinants of FL, as well as in studies that have analysed PHI, although none have analysed the possible moderating effect of age on the relationship between FL and PHI.

Regarding FL, Lusardi et al. [12] argue that FL changes over a person’s lifetime. Several studies have shown a lower level of FL among the youngest and oldest individuals (e.g., [12,40]). Atkinson and Messy [41] find that middle-aged individuals show higher levels of FL in most of the countries analysed, while older and younger individuals are more likely to have lower FL levels. There is a large body of literature showing that many elderly people have difficulty understanding the basics or the features of supplemental insurance [26].

According to Zheng [42]: “Age is likely to be a key factor in decisions on health insurance demand because the elderly population is generally exposed to more health risks and higher medical expenditures than the non-elderly population” (p. 47). Regarding empirical evidence, Kiil [43], in her review of the determinants of PHI, concludes that in most studies and countries, the relationship between age and the likelihood of holding PHI is positive (e.g., [25,44,45]), or there is positive relationship until a given age that becomes negative or insignificant thereafter (e.g., [46])—all referring to Spain. However, some studies also found a negative relationship between both variables in some countries, such as France (e.g., [47]).

Therefore, age is an important variable measuring one determinant affecting the probability of certain individuals seeking private healthcare. However, insurance companies go through a process of risk selection, whereby some individuals are excluded from private healthcare coverage due to adverse selection problems—for example, the very elderly—[24]. For this reason, health insurance premiums tend to increase with age, especially from 60–65 years of age, mainly due to the likelihood of pre-existing or underlying pathologies.

Hence, in relation to the elderly, there are two opposing effects. On the one hand, major health problems can motivate one to hold a health insurance policy. On the other hand, the higher cost of insurance can discourage such an acquisition. Hence, the net effect depends on which of the effects dominates. Thus, the second and last hypothesis is as follows:

**H2**.*Age moderates the relationship between FK and PHI*.

**H2a**.*Age moderates the relationship between basic FK and PHI*.

**H2b**.*Age moderates the relationship between advanced FK and PHI*.

## 4. Materials and Methods

### 4.1. Sample and Sources

The Financial Competencies Survey (ECF) is the source of information used in this study. This survey was carried out by the Bank of Spain in 2016 and it is the only one available to date. The sample was drawn by the National Statistical Institute and is representative of the population aged between 18 and 79, living in private households in Spain. It is also representative of each of Spain’s 17 autonomous regions [48]. The survey measures Spanish adults’ knowledge and understanding of financial concepts, as well as demographic variables and income level. The ECF contains responses from 8554 individuals between 18 and 79 years old. However, we excluded 499 observations with missing data for some of the variables. Since the income variable had 832 missing values, we completed this variable with the data extracted from the imputation provides for the Bank of Spain (see [48]). The final sample had 8055 observations. This survey is the only one that provides the information necessary to analyse the influence of FK on the frequency and type of PHI policies obtained by households in Spain.

### 4.2. Variables

#### 4.2.1. Dependent Variable

The only dependent variable was private health insurance (PHI), which was a dummy variable that adopted the value 1 if the respondent asserted that he/she had taken out PHI and 0 otherwise (e.g., [4,8]). This was based on the responses to the b0310 ECF question, which asks: *“Do you currently have subscribed, personally or jointly, any medical insurance? We refer to having it as a policyholder, not just as a beneficiary.”*

#### 4.2.2. Explanatory and Moderating Variables

According to the hypotheses, three explanatory variables about FK were created. As we have already mentioned, among the studies that consider FK as a proxy for FL are Lusardi and Mitchell [11,12], van Rooij et al. [13], Cupák et al. [14], Klapper et al. [15], and Klapper and Lusardi [16]. Specifically, the area of healthcare and FK as it relates to PHI, even when the FK consists only of basic numeracy, is considered by Fang et al. [26], Greene and Peters [27] and Paccagnella et al. [8].

The first variable is the financial knowledge index (FKI), which was created from the OECD/INFE [49] methodology and Engels et al. [19]. It is composed of seven items regarding numeracy, inflation, simple and compound interest, the risk-return relationship, diversification of investment products and mortgage term interest paid. The index was created by adding a point if the respondent answered each of these questions correctly, and zero in the cases where the answer was incorrect or if they answered “don’t know” [13]. Therefore, a higher value of the index represented a higher level of FK. The second and third variables were created by splitting FK into two levels, namely basic financial knowledge (Basic FK) and advanced financial knowledge (Advanced FK). Among the authors that have considered these types of FK are van Rooij et al., (2012) [17], Bannier and Schwarz [18], Engels et al. [19] and Gerrans et al. [20]. To create these indicators, items that corresponded to basic concepts were distinguished from those that represented more sophisticated financial concepts. Basic FK was computed for respondents regarding numeracy, inflation, and simple and compound interest. Advanced FK was considered as an understanding of the risk-return relationship, diversification of investment products and mortgage term interest paid. A description of variables can be seen in Table A1 of the Appendix A.

To contrast the moderating effect of the age, we considered two age groups: younger adults, under 50 years (Young-adult) and middle-aged and older, considered to be 50 years or more (Middle-older). Among the studies that have considered 50 years as the line between older and younger groups are Christelis et al. [50], Fong et al. [51], Bialowolski et al. [52] and Gerrans et al. [20]. Additionally, this allowed comparisons with the results of Paccagnella et al. [8], the only other study that has analysed the relationship between FK (albeit basic numeracy) and PHI in European countries, based on only approximately 50 people.

#### 4.2.3. Control Variables

Following the literature (e.g., [4,8,26,53]), several demographic and socio-economic variables were considered as control variables. Specifically, age, gender, level of education, marital status, children, income, occupation and location were included. Age was inserted into the models as a continuous variable (in logarithm). Gender was represented by the Women variable, which adopted the value 1 if the respondent was a woman, and 0 if it was a man. Marital status was a dummy variable that adopted the value 1 if the respondent was married (or cohabitating with a partner) and 0 if they were single. Children was a dummy variable that took the value 1 if the family had children and 0 if there were no children in the household. Level of education was collected through five dichotomous variables: primary, secondary, vocational training, baccalaureate and higher studies (Primary_studies (reference), Secondary_studies, Vocational_training, Baccalaureate, and Higher_education), adopting the value 1 depending on the highest educational level achieved by the respondent and 0 otherwise. Regarding occupation, five dichotomous variables were created: Unemployed (reference), Self-employed, Employee, Retired, and Inactive. The level of family income was introduced through six dichotomous variables, namely: IncomeN (N: from 1 to 6), one for each level of household income provided by the survey: below €9000 (reference), 9001–14,500, 14,501–26,000, 26,001–44,500, 44,501–67,500 and over €67,500. Lastly, location was determined through 17 dichotomous variables that represented geographical regions, one for each autonomous community.

#### 4.2.4. Analytical Strategy

First, a descriptive analysis was conducted to identify the sample characteristics. Then, chi-square and t tests were conducted to examine the characteristics of respondents holding PHI policies. Second, to contrast the hypotheses, we used a binary Probit regression model (e.g., [8]), with holding PHI as the dependent variable and total, basic and advanced FK as explanatory variables. Demographic and socio-economic variables, as well as location, were used as control variables. The Model 1 specification was as follows:*Pr (PHI) = β_0_ + β_1_ FK + β_2_ Women + β_3_ Agei + β_4_ Married + β_5_ Children + β_6_ Secondary_studies + β_7_ Baccalaureate + β_8_ Vocational_training + β_9_ Higher_education + β_10_ Self_employed + β_11_Employee + β_12_Retired + β_13_Other_inactive + β_14–18_ Income + β_19–33_ Region*

In Model 2, FK is replaced by basic FK and advanced FK. In this type of non-linear model, marginal effects should be analysed. The estimation of the models was carried out considering the weights provided by the ECF. For purposes of comparison, in the econometric analysis, the FK variables were normalised, subtracting the minimum and dividing by the range (max-min). For robustness, and with the purpose of discovering whether the unbalanced data regarding having PHI altered the initial results, we re-estimated the models, applying a resampling bootstrap option in Probit. All statistics and estimations were obtained using the econometric package Stata14.

## 5. Results

### 5.1. Descriptive Analysis

First of all, we must point out that according to the ECF-2016, the percentage of households in which the respondent stated that they had PHI was 20.43%. These data are consistent with the figures provided by the insurance employer in Spain, UNESPA [54], according to which 20.6% of the Spanish population was covered by PHI in 2016. Regarding age, the sample was made up of 44.38% of households whose head of family was 50 or over, compared to 55.62% where they were under 50 years of age. These proportions correspond to the population pyramid of Spain as of 1 January 2016, according to information extracted from the National Institute of Statistics [55] (INE). Specifically, considering the population between 20 and 79 years (according to the ECF), the percentage of individuals between 20 and 49 years is 56.74%, while the percentage of people in this population who are between 50 and 79 years is 43.26%. Therefore, the sample is considered representative of the Spanish population and of the population with health insurance coverage.

Table 1 shows the descriptive statistics of total, basic and advanced FK for all samples, as well as by PHI status and age group, specifically younger adults (Young-adult) and middle-aged and older (Middle-older).

As we can see, total, basic and advanced FK is higher in the PHI status than non-PHI, both in the Young-adult and Middle-older groups, and the differences are statistically significant. Moreover, in all FK variables, the figures for Middle-older are higher than for Young-adult when these have a PHI policy. On the contrary, these figures are lower when they don´t have PHI. These preliminary results support the hypotheses.

The descriptions of the moderator and control variables by PHI status are shown in Table 2. As we can see in this table, the subsample of individuals with PHI is made up of 60% Young-adult and 40% Middle-older. In the non-PHI subsample, the distribution is 55% Young-adult to 45% Middle-older. In both age groups, the difference between those with and those without PHI is statistically significant.

Regarding the control variables, all differences between PHI and non-PHI are also statistically significant, except for Women and Baccalaureate. The profile of people holding PHI policies are slightly younger, they are married or with a long-term partner (Married/partner), have children at home (Children), have a higher level of education, are self-employed or an employee or have a higher level of income.

Table A2 of the Appendix A collected the distribution of the sample for region (Autonomous communities) by PHI status. As we can see, three regions (Andalusia, Catalonia and Madrid) jointly represent more than 30% of the sample and near to 40% of the total PHI. In addition, these regions, along with another two (Canary Island and Basque Country), have a similar or higher value of the mean (20%) in individuals holding PHI with respect to their own regions. The variable is relevant because the Spanish healthcare system is regionally decentralised, and it is possible that some differences are due to political reasons. Among the studies on PHI that have taken region into account is that of Chatterjee and Nielsen [4].

Finally, the correlation matrix and variance inflation factor (VIF) can be seen in Table A3 of Appendix A. As we can see, all coefficients are lower than 0.40, except the correlation of FK with basic or advanced FK (0.74 and 0.86, respectively)—which is logical, since FK is the sum of the other two—and the correlation expected between Married/partner and Children (0.44). In addition, the correlation between basic FK and advanced FK is 0.29, so they can be included in the same econometric model. Likewise, the VIF is lower than 3. Thus, there should be no multicollinearity problems.

### 5.2. Econometric Results

The results of the Probit estimation are shown in Table 3. Models 1 and 2 are estimated over the whole sample, while Models 3 to 6 are estimated for age subsamples. Models 3 and 5 are estimated for Young-adult and Models 4 and 6 for Middle-older. The results of Model 1 indicate that the FK is positive and significant for the whole sample. In Model 2, we replaced total FK with basic and advanced FK. As can be seen in Table 3, both levels of FK are positive, but basic FK is not significant, while advanced FK is significant at 5%.

For age groups, the results of Models 3 and 4 indicate that the FK is positive, but it is only significant in Model 6 (at 1%), referring to Middle-older, but not to the Young-adult sample. In Models 5 and 6, we again replaced the total FK with basic and advanced FK. As we can see, in Model 5, for the Young-adult sample, none of FK variables are significant, while in Model 6, for the Middle-older sample, only advanced FK is significant (at 5%). These results support hypothesis H1, which predicts a positive relationship between FK and PHI, and also supports H1b regarding advanced FK, but not H1a regarding basic FK. Likewise, the differences between the age groups support the H2 and H2b hypotheses, but not H2a.

Regarding the control variables, the results of Models 1 to 4 (whole sample) are similar in sign and significance. Among demographic variables, only Women is significant (at 5% or 10%). Age (Age), being married or living with a partner (Married/partner) and having children at home (Children) are not significant. However, the level of education (Secondary studies, Vocational training, Baccalaureate, Higher education) and Occupation variables (compared to unemployment) are significant at 1% (5% for Inactive) in all models. Compared to the lower level (<9000 euros), all the income variables are positive and significant, except Income2 (between 9000 and 14,500 euros) in all models, except in Model 3 regarding the Young-adult group. In this model, the income levels 5 and 6 are not significant. Hence, there is an increasing relationship between income and PHI, especially in the Middle-older sample. In the Young-adult subsample (Models 5 and 7), the results of the control variables are similar to the whole sample, with the exception that Women and Retired (as expected) lose their significance.

In the subsamples of Middle-older (Models 6 and 8), the Women and Retired variables are again significant (both at 5%). The main difference is the significance of the Children variable in the older age models. In Models 5 to 8, Secondary studies is significant, albeit at 5%. In addition, in all models, almost half of the regions have a significant difference in respect to Andalusia, used as reference, although some of them have a positive sign whereas others have negative ones. This reveals some differences between the regions regarding the likelihood of holding PHI. In summary, gender is not significant in younger adults, and having children at home is only negative and significant in middle-aged and older people. The rest of the variables produced similar results in all models. Thus, a higher level of education and income increases the likelihood of holding PHI. Likewise, all occupations improve levels of PHI, in contrast to being unemployed. By contrast, age and being married or living with a partner did not have relevance with regard to the likelihood of holding of a PHI policy in Spain.

### 5.3. Robustness Analysis

Firstly, we considered three levels of FK: new basic, intermediate and advanced FK. The original basic FK was split into two variables. The new basic FK was integrated with numeracy and simple interest, and intermediate FK with compound interest and inflation. Advanced FK remained the original composition. The results of re-estimation are shown in Table 4. As we can see, in Model 7, regarding the whole sample, new basic and advanced FK are positive and significant at 5%, but intermediate FK is not. However, in the Young-adult sample, none of the FK variables are significant, and only advanced FK is significant (at 5%) in the Middle-older sample. Hence, the results are similar to those obtained in the Models 4 and 6.

Secondly, as indicated in the descriptive analysis, 20% of the sample had PHI, hence the sample was unbalanced. With the aim of considering whether this issue affected the results, we re-estimated the models by applying the resampling method, which is a specific bootstrap option in Probit. The resampling estimation provided results following different processes of respondent random selection. The results for the main variables are shown in Table 5. In this table, we present the results in two panels, namely A and B. In each panel, the result of the models for the whole sample and the two subsamples are shown. In panel A, we show the results for the models when the explanatory variable was FK and, in Panel B, when we considered basic and advanced FK as explanatory variables. These results were similar in sign and significance to those obtained in the initial models shown in Table 3.

Furthermore, in order to verify that the lack of significance of FK in the Young-adult subsample was not due to an aggregation effect, we re-estimated Models 3 and 5, splitting the sample into young people (<30 years old) and adults (30–49 years). The results (not reported for brevity) are similar in sign and statistical significance to those obtained in the model. In addition, given that new basic FK was significant in Model 7 and was not significant in any of the subsamples, we re-estimated this model for different age groups. The results indicate that it was significant and positive in the group under 30 years of age, but not in those aged 30–49, nor in those aged 50 or over. This may explain the significance of this variable in Model 7. On the other hand, it is worth noting that there were 15 retired people under 50 years old. This was because they had taken early retirement due to incapacity for work. Although this was a small number, we re-estimated Models 3 and 5 by excluding those who were retired. The results (non-reported) were maintained. Another variable that can affect the probability of people having PHI is the economic level of the region. To analyse this effect, we replaced the dummy variables representing the regions with GDP per capita (in logarithms). The estimation results (un-reported) indicated that GDP was positive and significant (at 1%), although the results related to the variables of interest were not altered.

In addition, some previous studies that have analysed the incidence of FL on other financial decisions, such as holding a pension plan (e.g., [56]), have considered the potential endogeneity of FL. To address this, following Fong and Straughan [57], we proceeded to evaluate the endogeneity of the FK variables. According to Cameron and Trivedy [58]: “…a regressor is endogenous when it is correlated with the error term” (p. 92). Thus, in order to ascertain whether the variables used as FK—basic and advanced—were endogenous, we estimated the residuals of the corresponding regressions and computed the correlation of those residuals and the FK variables. These correlations turned out to be insignificant in all the models (*p*-values > 0.80), so we ruled out the existence of endogeneity. Along this line, other studies on FK and PHI ([8,26,27]) have not considered the potential endogeneity either. In the same vein, Fong and Straughan [57], in their study about the relationship between FL and private long-term care insurance, which is similar to PHI, assert: “We conclude that there is no evidence that financial knowledge must be treated as endogenous rather than exogenous in our data” (p. 14).

## 6. Discussion

In this work, we analysed the relationship between possessing FK—total, basic and advanced—and the likelihood of holding PHI, as well as the moderating effect of age in this relationship. The results obtained from a sample of 8055 Spanish people indicate that the level of FK increases the likelihood of holding PHI but is only significant in the whole sample and for middle-aged and older people—not for younger adults. In addition, when we analysed two levels of FK, only advanced FK was significant. These results allow us to assert that there is a positive relationship between possessing advanced FK and holding PHI, as predicted in hypothesis 1. Likewise, the results are in line with hypothesis H1b regarding advanced FK, but not H1a, which references basic FK. We can assert that there are differences between the two levels of FK; hence, it is important to take this into consideration.

Regarding the age groups, the results indicate that none of the FK variables—total, basic or advanced—were significant for younger-adults, whereas only advanced FK was relevant to explaining the likelihood of middle-aged and older people holding PHI. Thus, we can assert that (1) basic FK is not relevant; and (2) in older people, only advanced FK contributes to the likelihood of holding PHI. Thus, we conclude that basic FK is not sufficient for these people; the ability to understand more complex financial concepts is what increases the likelihood of holding PHI in older age. These results show a difference between the two age groups, which proves that age has a moderator effect on the relationship between basic FK and holding PHI policy. The results offer support to hypotheses H2, H2a and H2b.

The results obtained in the present work have not been identified in previous studies. As indicated in the introduction, the only works that have analysed the relationship be-tween FK and PHI have been limited to numeracy (a component of basic FK) and middle-aged and older people. Hence, they are only comparable with our results regarding basic FK and the middle-older subsample. In this sense, the results obtained in our work differ from those obtained by Fang et al. [26] regarding the USA, and by Paccagnella et al. [8] for European countries, since basic FK was not significant in our study. This difference reveals the importance of distinguishing between basic and advanced FK. We must point out that other studies have considered samples with younger people, but they are not comparable with ours. Greene and Peters [27] consider individuals aged 18 or older, but use a subjective numeracy measure—not an objective one—and their study is based on a small convenience sample of 122 individuals. Likewise, Chatterjee and Nielsen [4] based their work on a sample from the USA of people aged from 37 to 49 years, but it considered cognitive ability, not FK.

The results regarding the importance of advanced FK for middle-aged and older peo-ple can be explained by the fact that PHI contracts for older people tend to be more com-plex due to the associated policies being more expensive, and the insurance companies include numerous clauses in order to restrict the use (or abuse) of healthcare services. The explanation could be that with aging, illnesses are more likely to arise, which could re-quire more frequent use of services covered by insurance. For the same reason, the absence of significance of FK in the Young-adult subsample can be explained by the fact that the PHI policies for these people have lower prices. Moreover, the health questionnaires given at the time of signing a contract are less demanding, presumably because insurance companies understand the lower risk associated with this group.

Regarding the control variables, it should be noted that the results obtained for level of education and income are in line with previous studies (e.g., [8,9,59]). The lack of a significant relationship between health insurance and high levels of income for the Young-adult group can be understood because, along with the lower probability of falling ill at those ages, a high income also reduces concern about health spending in the event of contracting an illness. However, with the Middle-older subsample, it must be taken into account that private insurance represents an expense in and of itself, which is compensated to the extent that greater use is made of the medical services covered by the insurance. Furthermore, it is used more intensively in older people. Regarding occupation, it is generally found that unemployment reduces the probability of people holding PHI (e.g., [25,47]). Conversely, being self-employed was generally found to increase the likelihood of holding PHI (e.g., [25,59,60]). In relation to gender, the empirical evidence is mixed. Jofre-Bonet [25] and Rodríguez and Stoyanova [59] found that gender is not a significant determinant of PHI coverage in Spain. However, other studies have found that men are more likely to hold a PHI policy [60]. The non-significant result regarding the relevance of age to the likelihood of holding PHI can also be found for Spain in the study of Pinilla and González [31]. Regarding marital status, the evidence from Spain is conflicting. Jofre-Bonet [25] found a negative effect of living alone and Costa-Font and Jofre-Bonet [46] found that being married had a negative effect on the probability of holding PHI in Spain. However, Chatterjee and Nielsen [4] found a positive relationship. Regarding children, our results are in line with other studies that found no significant link (e.g., [61]). Lastly, regarding location, our results indicate that living in certain regions increases the probability of holding PHI in Spain, similarly to other studies (e.g., [9,46]). This result can be partially explained by the decentralisation of the Spanish healthcare system, which causes differences in healthcare management at the regional level, including the quality of services (e.g., waiting lists).

Regarding the role of PHI at the macroeconomic level, there is certain debate about the advantages and disadvantages of PHI as an alternative or complement to the NHS. In this vein, some experts maintain that this weakens public healthcare resources. However, others assert that the use of healthcare services by those insured by the NHS releases some public resources, which can be used for those who do not have PHI, increasing the quality of public healthcare (e.g., by reducing waiting lists). In this sense, several studies show that double cover causes an increase in the utilisation of PHI, which may take some pressure off of public providers [31]. Following this idea, Cantarero-Prieto et al. [62], based on the data from the National Health Survey, suggests that Spanish people with PHI use the public healthcare system less than individuals without double health insurance coverage.

The results have several practical implications. On the one hand, for the policy maker, it is important to adopt measures to reduce waiting lists for public healthcare, which would contribute towards reducing spending on PHI. On the other hand, according to Paccagnella et al. [8], insurance companies should avoid the use of clauses that are difficult to understand, especially for the elderly. In addition, these firms could follow the advice of the supervisory board on selling PHI that gauges financial knowledge.

Finally, the main limitation of this work is the lack of a data panel, since the ECF microdata are currently only available for one year. This prevents us from considering a data panel that would avoid the unobservable heterogeneity bias. Possible extensions of the work include considering the type of PHI (co-payment, etc.), as well as the premium paid by the insurance. Likewise, the service offered by insurance companies in different regions could be considered.

## 7. Conclusions

We conclude that FK is relevant to holding PHI; specifically, advanced FK for middle-aged and older people in order to improve their health and financial wellbeing. In this sense, the increase in life expectancy and the consequent aging of the population should motivate governments to take the necessary measures to provide the elderly with advanced financial knowledge. On the other hand, governments must regulate the conditions that insurers establish in the policies of the elderly, eliminating certain abusive practices.

## Figures and Tables

**Table 1 healthcare-11-02738-t001:** Financial knowledge, Basic FK and Advanced FK by PHI and age.

Variable	Whole	PHI	Non PHI	*t*-Test
Financial knowledge (FK)				
Whole	4.87	5.23	4.78	−11.42 ***
Young-adult	4.94	5.16	4.88	−5.86 ***
Middle-older	5.79	5.33	4.67	−10.27 ***
Basic FK				
Whole	2.88	3.04	2.84	−7.93 ***
Young-adult	2.91	3.00	2.89	−3.69 ***
Middle-older	2.86	3.10	2.80	−7.57 ***
Advanced FK				
Whole	1.98	2.19	1.93	−10.36 ***
Young-adult	2.03	2.16	1.99	−5.55 ***
Middle-older	1.94	2.23	1.87	−9.11 ***

Average values. ***: significant at 1%. Source: own elaboration from ECF.

**Table 2 healthcare-11-02738-t002:** Average values of the moderator and control variables by PHI.

Variable	Whole	PHI	Non-PHI	Test
Young-adult	0.5562	0.5972	0.5456	14.10 ***
Middle-older	0.4438	0.4028	0.4544	14.10 ***
Age (years)	47.05	45.92	47.33	3.30 ***
Women	0.5028	0.4945	0.5049	0.56
Married	0.6606	0.7102	0.6478	22.72 ***
Children	0.4792	0.5055	0.4725	5.71 **
Secondary Studies	0.2694	0.1829	0.2916	78.70 ***
Baccalaureate	0.2148	0.2157	0.2145	0.01
Vocational Training	0.1261	0.1464	0.1209	7.72 ***
Higher education	0.2340	0.4046	0.1902	335.89 ***
Self-employed	0.1153	0.1883	0.0966	108.06 ***
Employee	0.4282	0.4848	0.4136	27.10 ***
Retired	0.1549	0.1306	0.1612	9.34 ***
Other inactive	0.1641	0.1258	0.1740	22.19 ***
Income2 (9.001–14.500)	0.2133	0.1258	0.2358	94.46 ***
Income2 (14.501–26.000)	0.2837	0.2375	0.2955	21.66 ***
Income4 (26.001–44.500)	0.2323	0.3056	0.2134	62.35 ***
Income5 (44.501–67.500)	0.0937	0.1707	0.0740	144.34 ***
Income6 (>67.000 €)	0.0473	0.1075	0.0318	166.56 ***
Total observations	8055	1646	6409	

See variable descriptions in Table A1 of Appendix A. Values computed one for dichotomous variables and over total observations. Test: *t*-test in continuous variables and Chi2 in dummies, for difference of means. In both cases, a comparison is made between those PHI and non-PHI. *** and **: significant at 1% and 5%, respectively. Source: own elaboration from ECF.

**Table 3 healthcare-11-02738-t003:** Financial knowledge and PHI.

Model	Model 1		Model 2		Model 3		Model 4		Model 5		Model 6	
Sample	Whole		Whole		Young-Adult		Middle-Older		Young-Adult		Middle-Older	
	M.E.	S.E.	M.E.	S.E.	M.E.	S.E.	M.E.	S.E.	M.E.	S.E.	M.E.	S.E.
Financial Knowledge	0.0323 **	0.0133	-	-	0.0108	0.0190	0.0525 ***	0.0177	-	-	-	-
Basic FK	-	-	0.0099	0.0248	-	-	-	-	0.0331	0.0344	0.0535	0.0338
Advanced FK	-	-	0.046 **	0.0182	-	-	-	-	0.0365	0.0256	0.0518 **	0.0252
Age (log)	−0.0097	0.0185	−0.0093	0.0185	−0.0454	0.0333	−0.0032	0.0855	−0.0432	0.0333	−0.0031	0.0854
Women	0.0235 **	0.0105	0.0232 **	0.0105	0.015	0.0143	0.0371 **	0.0152	0.014	0.0143	0.0371 **	0.0152
Married/partner	0.0170	0.0128	0.017	0.0127	0.0253	0.0199	0.0069	0.0172	0.0248	0.0198	0.0069	0.0172
Children	−0.0115	0.0114	−0.0118	0.0114	0.0202	0.0199	−0.0425 ***	0.0152	0.0195	0.0199	−0.0424 ***	0.0151
Secondary studies	0.0850 ***	0.0225	0.0853 ***	0.0226	0.1018 **	0.0424	0.0518 **	0.0245	0.1022 **	0.0426	0.0518 **	0.0245
Baccalaureate	0.1350 ***	0.0262	0.1358 ***	0.0263	0.1036 ***	0.0421	0.153 ***	0.0354	0.105 ***	0.0424	0.1529 ***	0.0354
Vocational Training	0.1755 ***	0.0386	0.1761 ***	0.0309	0.1346 ***	0.0472	0.2157 ***	0.0465	0.1357 ***	0.0474	0.2157 ***	0.0465
Higher education	0.2131 ***	0.0278	0.2143 ***	0.0279	0.1563 ***	0.0447	0.2762 ***	0.0381	0.1588 ***	0.0450	0.2761 ***	0.0381
Self-employed	0.1831 ***	0.0282	0.1842 ***	0.0282	0.2028 ***	0.0356	0.1754 ***	0.0471	0.2043 ***	0.0356	0.1753 ***	0.0471
Employee	0.0566 ***	0.0182	0.0572 ***	0.0183	0.0578 ***	0.0220	0.0662 **	0.0330	0.0592 ***	0.0219	0.0662 **	0.0329
Retired	0.0677 ***	0.0273	0.0879 ***	0.0273	0.0391	0.1505	0.0794 **	0.0359	0.0289	0.1464	0.0794 **	0.0358
Other inactive	0.0540 **	0.0231	0.0549 **	0.0232	0.0616 **	0.0311	0.0579 *	0.0379	0.065 **	0.0312	0.0579 *	0.0377
Incomes 2	0.0280	0.0221	0.0281	0.0221	0.0295	0.0308	0.0287	0.0305	0.0294	0.0308	0.0287	0.0305
Incomes 3	0.0595 **	0.0216	0.0596 ***	0.0217	0.0754 ***	0.0302	0.0492 *	0.0302	0.0758 ***	0.0302	0.0492 *	0.0301
Incomes 4	0.1267 ***	0.0245	0.1269 ***	0.0245	0.1603 ***	0.0335	0.1009 ***	0.0357	0.1608 ***	0.0335	0.1009 ***	0.0356
Incomes 5	0.2141 **	0.0332	0.2138 ***	0.0332	0.2627	0.0448	0.1608 ***	0.0486	0.2624 ***	0.0448	0.1608 ***	0.0486
Incomes 6	0.2691 ***	0.0399	0.2699 ***	0.0400	0.3636	0.0545	0.1696 ***	0.0551	0.3659 ***	0.0546	0.1696 ***	0.0550
AACC (Region)	Yes		Yes		Yes		Yes		Yes		Yes	
Observations	8055		8055		4480		3575		4480		3575	
Pseudo-R^2^	0.1083		0.1085		0.0953		0.1497		0.0959		0.1497	
Log-likelihood	−3630.95		−3630.21		−2124.30		−1460.24		−2122.91		−1460.23	

Dependent variable: PHI, binary = 1 if the respondent has a PHI and 0 if not. See variable descriptions in Table A1 of Appendix A. Estimation method: Probit. M.E.: Marginal effects; S.E.: Standard errors. ***, **, *: significant at 1%, 5% and 10%, respectively. Source: own elaboration from ECF.

**Table 4 healthcare-11-02738-t004:** Financial Knowledge and PHI. Robustness (I).

Sample	Whole		Young-Adult		Middle-Older	
	M.E.	S.E.	M.E.	S.E.	M.E.	S.E.
	Model 7		Model 8		Model 9	
New basic FK	0.0384 **	0.0168	0.0350	0.0232	0.0381	0.0234
Intermediate FK	0.0072	0.0144	0.0281	0.0193	0.0171	0.0211
Advanced FK	0.0414 **	0.0185	0.0315	0.0259	0.0484 **	0.0256
Control variables	Yes		Yes		Yes	
P-seudo R^2^	0.1093		0.0968		0.1502	
Log-likelihood	−3626.86		−2120.74		−1459.54	

Dependent variable: PHI, binary = 1 if the respondent has a PHI and 0 if not. See variable descriptions in Table A1 of Appendix A, Estimation method: Probit. M.E.: marginal effects. S.E.: Standard errors. **: significant at 5%. Source: own elaboration from ECF.

**Table 5 healthcare-11-02738-t005:** Financial knowledge, age and PHI. Robustness (II).

Sample	Whole		Young-Adult		Middle-Older	
	M.E.	S.E.	M.E.	S.E.	M.E.	S.E.
*Panel A. FK*						
	Model A1		Model A2		Model A3	
Financial knowledge	0.0231 **	0.0392	0.0175	0.0571	0.1799 ***	0.0699
Control variables	Yes		Yes		Yes	
Pseudo-R^2^	0.1144		0.0957		0.1571	
Log-likelihood	−3612.07		−2131.71		−1445.09	
*Panel B. Basic and advanced FK*						
	Model B1		Model B2		Model B3	
Basic FK	0.0105	0.0175	0.045	0.0349	0.0272	0.0271
Advanced FK	0.0437 **	0.0136	0.1226	0.0786	0.0500 **	0.0199
Control variables	Yes		Yes		Yes	
Pseudo-R^2^	0.1148		0.0964		0.1572	
Log-likelihood	−3610.39		−2129.92		−1444.94	
Observations	8055		4480		3575	

Dependent variable: PHI, binary = 1 if the respondent has a PHI and 0 if not. See variable descriptions in Table A1 of Appendix A. Estimation method: Probit with bootstrap. M.E.: Marginal effects. S.E.: Standard errors. *** and **: significant at 1% and 5%, respectively. Source: own elaboration from ECF.

## Data Availability

The dataset used from the Financial Competences Survey (ECF) for this study is publicly available at: Bank of Spain, Survey of Financial Competences, Website of Bank of Spain. https://www.bde.es/bde/es/areas/estadis/Otras_estadistic/encuesta-de-comp/ (accessed on 19 June 2023).

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
