# Peer review of "Financial Knowledge and Private Health Insurance: Does Age Matter?"

_healthcare, 2023, doi:10.3390/healthcare11202738_

Round 1
Reviewer 1 Report
This article analyses the relationship between Financial Knowledge and the likelihood of paying and holding a Private Health Insurance. Curiously enough, the moderating variable is age and not income, as the latter could be more important or significant. Furthermore, there is a relation between age and income that the article should clarify. In fact, in Spain the risk of poverty is much higher at lower ages than elder people. Last average retirement pensions are higher than average wages and much higher than younger age salaries. The article ought to quantify or illustrate about the relation between age and income. Obtaining that the level of Financial Knowledge is positively related to holding a Private Health Insurance can hide this circular relation, especially when it is not significant in the whole sample and only for middle-aged and older people (not for younger adults). Some studies have found that men are more likely to hold a PHI policy for the same reason.
English language is fine, but there is an acronym that it is not defined. It is FL. Does it stand out for Financial Literacy? It appears in many places along the article:
. FL can be defined as...
. Lusardi et al. argue that FL changes over a person's lifetime.
. Specifically, FL is critical among certaing groups...
. The literature offers several definitions of FL.
...
English language is fine, but there is an acronym that it is not defined. It is FL. Does it stand out for Financial Literacy? It appears in many places along the article:
. FL can be defined as...
. Lusardi et al. argue that FL changes over a person's lifetime.
. Specifically, FL is critical among certaing groups...
. The literature offers several definitions of FL.
...
Author Response
First of all, we want to thank you for your comments and the time dedicated to reviewing our work. Below we indicate the response to your suggestions:
1. Following your suggestion, we have expanded the text corresponding to the explanation of income as a control variable, on page 11 (lines 421-424), as well as in the discussion of results (pages 13-14, lines 532-540)
In addition, we have proceeded to create an interaction variable between age and income. Specifically, we have followed Reviewer 2's proposal regarding considering income as a categorical variable that takes values from 1 to 6, instead of dummy variables. We have re-estimated the models including this new variable, although the interaction is not significant. So, we conclude that the results are not affected by this issue.
2. We have proceeded to insert the acronym at the first place in the text that we cite ‘financial literacy’, on page one (line 25), and the concept is defined on page 2, lines 47-49. Likewise, to homogenise the text, we have replaced the phrase ‘financial literacy’ with ‘FL’ on pages 1 y 2, after having cited the acronym.
Reviewer 2 Report
Summary: This study seeks to distinguish the differential effects of financial knowledge on the propensity to hold a private health insurance, suggesting older aged people and more educated samples are inclined to buy more private health insurances.
Comments:
How would your results be affected if you consider three levels of financial literacy, namely, basic, intermediate, and advanced? This may serve as a robustness check.
Can you clarify whether in your young-adult samples, there are respondents who classify themselves as retirees? It seems “retired” is insignificant in models 3 or 5, but I am not sure if this is due to a dearth of data that may potentially bias the results.
Can you consider income and education level as categorial instead of classified dichotomous variables, and include the interaction of FK and age/education level?
Have you winsorized your data to remove the top/bottom 1% or 5% outliners?
I think they should highlight some policy implications in their discussions, such as perfecting the supervisory board on selling PHI that gauges financial knowledge.
The restriction of one-year data is too scarce to mention as limitations, and I think the authors should map out the future directions for research in one or two paragraphs.

I think the authors should double check their spelling prior to submission.
Author Response
First of all, we want to thank you for your comments and the time dedicated to reviewing our work. Below we indicate the response to your suggestions:
1.
Following your suggestion, we have proceeded to create two new variables labelled ‘new basic FK’ and ‘intermediate FK’, while ‘advanced knowledge’ has been maintained. To create the first two variables, we divided the original basic FK into two. Thus, the new basic FK is composed of numeracy and simple interest, and the intermediate of compounded interest and inflation. We have re-estimated models 2, 4 and 6 and the results are similar to those initially obtained, except for the new basic FK, which is significant in the model of the whole sample (model 7). The results are in a new table - Table 4 (see page 11).
2. In Spain, the presence of retirees in the Young-adult subsample occurs because some people have retired before the established age (around 65 years), generally motivated by a work incapacity. In our sample, there are 15 retired people under 50 years old.
In the robustness section, we have included a text explaining this question. Specifically, we comment that there are only 15 retired people in the Young-adult sample and that we have re-estimated the models, and the results are similar to the previous ones (see page 12, lines 466-469). Since there is hardly any difference in the sample, we considered the new results not interesting enough to report, but are available upon request.
3. Following your suggestion, we have created the categorical variables for income (values from 1 to 6) and education (values from 1 to 5). Likewise, we have created the interaction between age and income, as well as age and education. The results of the re-estimated models indicate that none of these interactions are significant. Therefore, we have not considered these results to be worthy of inclusion in the work, although they are available upon request.
4. We have not winsorized any variable as they are all dummy variables. The only exceptions are the financial knowledge variables (values in levels) and age, which is computed as a logarithm form.
5. Following your suggestion, we have inserted a new text regarding the policy implications (see page 14, lines 568-573), expressly including your suggestion. We greatly appreciate this idea.
6. Following your suggestion, we have inserted a new text regarding the future directions of research (see page 14, lines 576-579).
7. According your suggestion, we have proceeded to revise the text.
Reviewer 3 Report
Moderate revision is required

Minor editing is required
Author Response
First of all, we want to thank you for your comments and the time dedicated to reviewing our work. We have copied the text of your questions or suggestions from the pdf and we respond here to each of them:
- Response to Q1: We have proceeded to insert the acronym at the first place in the text that we cite the phrase ‘financial literacy’, on page one (line 25), and the concept is defined on page 2, lines 47-49. Likewise, to homogenise the text, we have replaced the expression ”financial literacy” with FL on pages 1 y 2, after having cited the acronym
-
Response to Q2. In the new version of the work we have changed “they” for “those results”
-
Response to Q3: We prefer to leave it even though it has been said in the introduction so that the reader can follow the thread of the text in the literature review section.
-
Response to Q4: We added “to”
-
Response Q5: Regarding the clarification about the g1110 ECF question, thanks to your observation, we have detected an error and we have proceeded to correct it. As we indicated in Tablel A1 of Appendix, the correct question is b0310 instead g1110, and we have added the literal text of the aforementioned question in the ECF (see page 6, lines 274-276, in new version).
-
Response to Q7: Once again, thanks to your observation, we have detected an error in the comment of the econometric results in page 10, specifically in the explanatory text of the results relating to model 3. We have proceeded to correct it. In the new version, the text coincides with the results that appear in Model 2, Table 3 (see page 10, lines 395-396 in new version).
-
Response to Q8: we have changed ‘PH’ for ‘PHI’
-
Response to Q9: It is table 3. In the new version, we have extended the explanation.
Round 2
Reviewer 2 Report
Why the new basic financial knowledge is significant for the whole sample at the 5% level but neither with young-adult and middle-older people in Table 4? Also note that the borderline at the leftmost column of last row is incomplete.
Line 470-1: simplify "due to being in a situation of" to "attributed to".
Line 546: "that use tends to be greater" -> "it is used more intensively"
Author Response
Response to reviewer 2 (second round)
Results Table 4
Regarding the results about the new basic FK, we have verified that the results have been transcribed correctly. Indeed, new basic FK is positive and significant in the overall sample, although it is not significant in either of the two subsamples. In order to analyze the reason for this result, we have reestimated the two subsamples for different age groups. I the new version of the work we have include this sentence in page 12):
"In addition, given that new basic FK is significant in model 7 and is not significant in any of the subsamples,
we have re-estimated this model for different age groups. The results indicate that it is significant and positive in the group under 30 years of age, but not in those aged 30-49 nor in those aged 50 or over.T his may explain the significance of this variable in model 7."
Quality og English language
We have replaced the text according your suggestion
Reviewer 3 Report
Thank you, I have checked it and found it almost alright. But I found that the authors did not find one article that I had recommended to include in the revised version. I have given the link below.
https://rsijournal.eu/?p=4322
Author Response
First of all, we apologize for not being able to locate the article, and thank you for sending us the link. We have downloaded the paper of Hussain et a. (2022) to which the link you sent us corresponds and you indicated to us in your previous review. We found the article very interesting to the extent that it relates knowledge with health. Consequently, we have decided to include a citation in the first paragraph of the introduction